# The Impact of Physical Activity on the Circadian System: Benefits for Health, Performance and Wellbeing

**Dietmar Weinert [1,\*] and Denis Gubin [2,3,4]**

1. Institute of Biology/Zoology, Martin Luther University, 06108 Halle-Wittenberg, Germany
2. Laboratory for Chronobiology and Chronomedicine, Research Institute of Biomedicine and Biomedical Technologies, Medical University, 625023 Tyumen, Russia
3. Department of Biology, Medical University, 625023 Tyumen, Russia
4. Tyumen Cardiology Research Center, Tomsk National Research Medical Center, Russian Academy of Science, 634009 Tomsk, Russia
\* Correspondence: dietmar.weinert@zoologie.uni-halle.de

**Featured Application: This review summarizes current knowledge regarding the effects of physical activity on the robustness of the circadian system and the benefits for health, performance, and wellbeing. We propagate practical implications of circadian activity rhythms for diagnosis and the advantages of scheduled motor activity.**

**Abstract:** Circadian rhythms are an inherent property of all living systems and an essential part of the external and internal temporal order. They enable organisms to be synchronized with their periodic environment and guarantee the optimal functioning of organisms. Any disturbances, so-called circadian disruptions, may have adverse consequences for health, physical and mental performance, and wellbeing. The environmental light–dark cycle is the main zeitgeber for circadian rhythms. Moreover, regular physical activity is most useful. Not only does it have general favorable effects on the cardiovascular system, the energy metabolism and mental health, for example, but it may also stabilize the circadian system via feedback effects on the suprachiasmatic nuclei (SCN), the main circadian pacemaker. Regular physical activity helps to maintain high-amplitude circadian rhythms, particularly of clock gene expression in the SCN. It promotes their entrainment to external periodicities and improves the internal synchronization of various circadian rhythms. This in turn promotes health and wellbeing. In experiments on Djungarian hamsters, voluntary access to a running wheel not only stabilized the circadian activity rhythm, but intensive wheel running even reestablished the rhythm in arrhythmic individuals. Moreover, their cognitive abilities were restored. Djungarian hamsters of the arrhythmic phenotype in which the SCN do not generate a circadian signal not only have a diminished cognitive performance, but their social memory is also compromised. Voluntary wheel running restored these abilities simultaneously with the reestablishment of the circadian activity rhythm. Intensively exercising Syrian hamsters are less anxious, more resilient to social defeat, and show less defensive/submissive behaviors, i.e., voluntary exercise may promote self-confidence. Similar effects were described for humans. The aim of the present paper is to summarize the current knowledge concerning the effects of physical activity on the stability of the circadian system and the corresponding consequences for physical and mental performance.

**Keywords:** motor activity; circadian rhythms; circadian disruption; health; wellbeing; physical and mental performance

## 1. Introduction

Circadian rhythms are an inherent property of all living systems and are generated by an internal clock. In mammals, this is localized in the suprachiasmatic nuclei (SCN) of the hypothalamus [1]. The SCN as central pacemakers are involved in the rhythmic regulation

of sleep–wake behavior [2], locomotor activity [3], metabolism [4], body temperature [5], and other body functions.

Circadian rhythms are an essential part of the external and internal temporal order. They enable organisms to be synchronized with their periodic environment, to be prepared for rather than react to periodic changes and coordinate physiological and behavioral processes. This guarantees an optimal functioning, and any disturbances, or so-called circadian disruptions, may have adverse consequences for health, physical and mental performance, wellbeing, and longevity [6–10]. Such disruptions do occur during aging [11–14], following time-zone transitions [15–17], in shift-workers [18,19], and in subjects suffering from different diseases [20–23]. However, in the latter case, it is not clear if circadian disruptions are a consequence of the disease or a cause.

Therefore, it is highly important to find approaches to synchronize circadian rhythms and, thus, stabilize the entire circadian system. Melatonin, as an effective chronobiotic, is often used [24]. However, the physiological effects of melatonin depend critically on dosing and timing. For chronobiotic purposes, physiologic doses (3 mg or less) are useful, while higher doses (3–10 mg or more) exert antioxidant and immunoregulatory effects [25,26]. Optimal dosing may be different between individuals. Even more important could be a personal adjustment of timing, since physiological responses to melatonin follow phase response curves [27]. When non-physiological doses have to be used, one must consider adverse side effects. Accordingly, non-pharmacological approaches may be preferred, except, for example, in very old people where the use of melatonin might be unavoidable [28,29].

The daily light–dark cycle is the main zeitgeber [30,31]. It synchronizes circadian rhythms with the 24-h environment. In our modern society, however, most people are exposed to only low levels of light in the daytime. On the contrary, the light levels in the evening or at night are often too high. Both have adverse consequences for the synchronization of the circadian rhythms with the 24-h environment and multiple negative effects on physiology, sleep, physical and mental performance, and well-being [19,32–35]. Accordingly, circadian light hygiene and a proper 24-h light regimen in the offices and homes become increasingly important [36–38].

Food can also serve as a synchronizer for the circadian system. Timed feeding may reset the SCN phase via a reinforcement of clock gene expression, entrain peripheral clocks, and facilitate intrinsic synchronization [39–42]. Thus, it can be used to strengthen the circadian system, which in turn promotes healthy aging and longevity [21,43,44].

Robust circadian clocks can also be facilitated by lifestyle, regular physical activity, and timed exercise. Aged people often adopt, although rather intuitively, a regular lifestyle [45,46]. However, a problem of our modern society is a general low level of physical activity [47]. This has adverse consequences for the cardiovascular system and the energy metabolism [48]. Moreover, direct effects on homeostatic physiological mechanisms and disturbances of the circadian system have a considerable impact. Scheduled physical activity is an efficient strategy to maintain circadian rhythms and contribute to healthy ageing [12,13,49]. It upholds feedback coupling with the central brain clock and helps to preserve synchronized circadian rhythms [10,50].

The aim of the present paper is to summarize the current knowledge concerning the effects of physical activity on the stability of the circadian system and the consequences of elevated daily activity for physical and mental health and performance. Most of the data came from experiments on animals. However, as they reflect general biological principles, they are equally true for humans.

## 2. Physical Activity and the Circadian System

### 2.1. The Diagnostic Value of Circadian Activity Rhythms

In humans, activity can be monitored easily over many days and at short sampling intervals by means of non-invasive devices worn on the wrist of the non-dominant hand [51]. This method is widely used in field research and in clinical settings, as it is simple, inexpensive, and rather accurate [52–54]. In laboratory animals, motor activity can be monitored by

transmitters that are implanted into the peritoneal cavity and allowing the simultaneous recording of core body temperature [55]. Moreover, most reliable and less expensive are passive infrared motion detectors that are mounted above the cage roof [56]. Both methods record general activity, not just locomotion.

The wide use of actimetry monitoring techniques makes circadian activity rhythm assessment a useful tool to survey circadian rhythm robustness [57,58]. Moreover, rhythm characteristics can be used as indicators of physical and mental fitness and the health status of human beings. Vitale and co-authors [59] used actigraphy to characterize the activity level and the daily activity rhythm of patients following hip and knee joint replacement and designed specific, personalized rehabilitation programs.

The inter-daily rhythm stability and the median level of daytime activity were found to be negatively correlated with cognitive decline and depression in elderly women [53]. Other authors compared the amounts of activity when in bed or out of bed. This can be a preferable option when the 24-h pattern is non-sinusoidal. The estimated dichotomy indices were decreased in diseased individuals, particularly in those suffering from colorectal cancer [54,60]. Together with the mean activity level and autocorrelation coefficients, they are an objective indicator of physical welfare and an appropriate prognostic factor for cancer patients´ survival and tumour response [61–63]. According to Hoopes et al., characteristics of the rest–activity rhythm, especially the inter-daily stability and intra-daily variability, can be used as biomarkers of cardiometabolic health [64]. Irregular sleep duration and timing were recognized as novel cardiovascular risk factors, independent of other traditional factors and sleep quantity or quality [65]. An increased fragmentation of the 24-h activity rhythm is also positively associated with symptoms of food addiction [66]. Differences in the period length of the free-running activity rhythm were found in spontaneous hypertensive rats as compared to healthy controls [23]. Remarkably, changes occurred in the pre-hypertensive state already and not after the surgical induction of hypertension. In glaucoma patients, the sleep–wake rhythm was found to be compromised [67].

In laboratory animals, the onset of their main activity period is a convenient phase marker. It characterizes the strength of photic entrainment and undergoes predictable changes, for example, following a phase shift of the light–dark cycle [55,68] or during aging [69,70]. In old mice, a phase advance and a decreased phase stability of the activity onset were observed together with an increased inter-daily rhythm variability. These changes were found to happen earlier than the decrease in total activity per day and can be used to characterize the biological age [71,72]. Interestingly, similar changes were observed for the sleep–wake rhythm of aged people, particularly those suffering from Alzheimer´s disease (AD) and Parkinson's disease [53,73–75]. In mild-to-moderate AD, increased fragmentation of the activity pattern predicted cognitive decline that was independent of age, sex, educational level, and season [76]. Interestingly, this work also reported an increased circadian amplitude, illustrating that rhythm stability, amplitude, and phase should be considered altogether to examine the distinctive types of circadian disruption, e.g., "extra-circadian dissemination" [57].

### 2.2. Beneficial Effects of Increasing the Daily Activity Level

Worldwide, a high percentage of people is physically inactive, especially in high-income countries, and this percentage increases with age [47,77]. The scarce physical activity of elderly people is at least partly caused by sarcopenia, though its progression can be reduced by regular exercise [78,79].

Physical inactivity is one of the most important factors causing lifestyle diseases. It substantially increases the risk of adverse health conditions and shortens life expectancy. Particularly, the incidence of coronary heart disease, metabolic diseases such as obesity and type 2 diabetes, and also breast and colon cancers has been shown to be elevated. Accordingly, a physically active behavior could improve health and has a protective effect on the development of diseases [48,80,81]. The improvement of methods to record physical

activity might help to develop programs to enhance activity levels and, thus, reduce the risk of non-communicable diseases.

Regular exercise helps to prevent cardiovascular disease [82], stroke [83], cancer [84], diabetes, and obesity [85]. It also has positive effects in patients suffering from neuropsychiatric disturbances. Aside from psychological mechanisms such as improved sleep and stress reduction, neurobiological mechanisms such as changes in neurotransmitter concentrations are also involved [86,87]. Exercise can prevent the development of stress-related mood disorders such as depression and anxiety and promote self-confidence. Animal models were used to elucidate the underlying mechanisms [88]. In Syrian hamsters, voluntary exercise promotes resilience to social defeat; animals are less anxious and show less defensive/submissive behaviors [89].

Physical activity positively influences cognitive function in patients with dementia, i.e., it decelerates the cognitive decline [90,91]. Experiments in rodents revealed that physical exercise regulates hippocampal neurogenesis, especially if it is performed in the context of cognitive challenges, and reduces learning deficits [92–96].

*2.3. Effects of Motor Activity on the Circadian System*

The above-mentioned effects of physical activity are at last in part mediated via the circadian system. It has been well known for many years that motor activity has a considerable impact on circadian rhythms, via a feedback effect on the central pacemaker, the SCN. A correlation between activity level and the period length of the rhythm generated in the SCN was found [56,97,98]. Intensive running wheel use also affects photic phase responses [99–101]. Bouts of physical activity may cause phase changes—so-called non-photic phase responses. Depending on the circadian time, activity bouts may cause a phase advance or a phase delay. Mrosovsky first published a non-photic phase response curve [102]. In Golden hamsters for example, a bout of motor activity in the middle of the light time, i.e., during the subjective day, causes a phase advance. The animals' next activity period starts earlier, i.e., the animals "wake up" and "go to sleep" earlier. Similar effects were described for humans [103]. One hour of moderate treadmill exercise for three consecutive days caused significant phase changes of the urinary aMT6s rhythm, the metabolic end-product of melatonin.

Motor activity and behavioral states may directly influence the neurophysiological properties of the SCN and affect clock genes expression [104–107]. This may be the causal reason for the above-mentioned changes in photic phase responses and free-running periods.

Steinlechner and co-authors [101] argued that the circadian rhythms of Djungarian hamsters can be stabilized by enhanced motor activity. Indeed, Ortega and coworkers demonstrated that wheel running rendered the synchronization of the circadian body temperature rhythm in the golden hamsters [108]. This hypothesis was further supported by studies on aging mice. When animals had access to running wheels, the activity onsets, which were highly variable and phase advanced, became stronger coupled to "lights-off" [14,109].

In Djungarian hamsters with a delayed activity onset, photic entrainment was stabilized by means of elevated motor activity [98,110]. Like many other rodent species, Djungarian hamsters use running wheels voluntarily, regularly, and intensively [111,112]. Some of them eventually developed a wild-type activity pattern with its onset being tightly coupled to "lights-off"; however, this phenomenon could not be explained by the observed changes of endogenous period lengths and photic phase responses. Obviously, mechanisms downstream from the suprachiasmatic nuclei are involved.

Circadian rhythms are reinstated, and cognitive performance is improved in arrhythmic (AR) Djungarian hamsters when they intensively use running wheels [10]. The underlying mechanisms of the AR phenomenon are not yet known. The loss of synchronization between single SCN neurons, similar to what was found in aged mice, is a putative factor [113]. In SCN slices of AR Djungarian hamsters a broad range of firing frequencies in the electrical activity was found [114]. The authors argued that this indicates a loss of

synchrony among SCN neurons. Our study showed that resynchronization and a proper phase relationship between SCN neurons can be achieved by motor activity. Regardless, a more robust output signal downstream from the SCN should foster consistent overt circadian rhythms such as activity.

The observed phenomenon might also be caused by another mechanism. In vivo studies demonstrated that behavioral activity may suppress the firing rate in the SCN of freely moving rodents, e.g., rats and hamsters [106,115], and mice [116]. Since behavioral activity can modulate the SCN firing rate, it may also boost circadian rhythms when activity occurs at the specific time (at night in the case of nocturnal species). These findings are supported by a study which showed that scheduled daily activity can improve behavioral circadian rhythms in mice with compromised SCN function [117]. The 24-h pattern of wheel-running activity observed in our study on AR Djungarian hamsters can also be recognized as some type of scheduled activity, since hamsters utilize the wheels only during the dark phase [10].

The enhancement of circadian rhythm amplitude in the SCN is beneficial for other rhythms controlled by the central pacemaker [116]. Interestingly, only voluntary exercise functions, as it is the arousal and namely the positive arousal that has an effect [118]. Aschoff called this "feedback for fun" (personal communication). This is also important for the beneficial effects, which are discussed in the next section. Physical exercise must be voluntary and fun, otherwise it is much less effective, if at all.

In humans, personalized timing, duration, and the type of exercise should enhance its benefits, particularly for cardiovascular health [119]. Evening exercise might be less beneficial [120]. Moreover, benefits may vary depending on chronotype [121,122], or when metabolic circadian rhythms are compromised in diseases such as diabetes [123]. Overall, considering the timing of exercise can be important in performance and disease contexts to improve circadian alignment and health status as a therapeutic and preventative strategy [124].

In humans, physical activity closely associates with time spent outdoors [125]. Time outdoors was a factor of higher physical activity during COVID-19 mandated lockdown [126]. It was also linked with better sleep, a higher quality of life, and less pronounced phase delay, which was common during lockdown for the majority of the population [126]. It this context, it should be outlined that outdoor activity is related to outdoor light exposure, and these factors act together in facilitating circadian alignment and robustness. Overall, more outdoor time can be linked with a more active lifestyle and lower chronic disease risks [125,126].

This review is mainly based on animal studies, limiting the immediate translation and application of the findings to humans. Therefore, future research aimed at clarifying the benefits of timed physical activity in men and women depending on age, chronotype, and relevant genetic polymorphisms are prompted.

### 2.4. Beneficial Effects of Stabilized Circadian Rhythms

As described above, physical activity has a strong impact on circadian rhythms, Figure 1. In rodents, it has been shown that voluntary access to a running wheel helps to maintain the high-amplitude circadian rhythms of clock and clock-controlled genes in the SCN. Similar effects can also be expected to occur in humans. As a consequence of high-amplitude circadian rhythms in the SCN, their rhythmic output that controls overt rhythms and peripheral oscillators is strengthened. Proper internal and external phase relationships can be established, and a circadian disruption be abolished [109,117]. This reduces the risk of diseases.

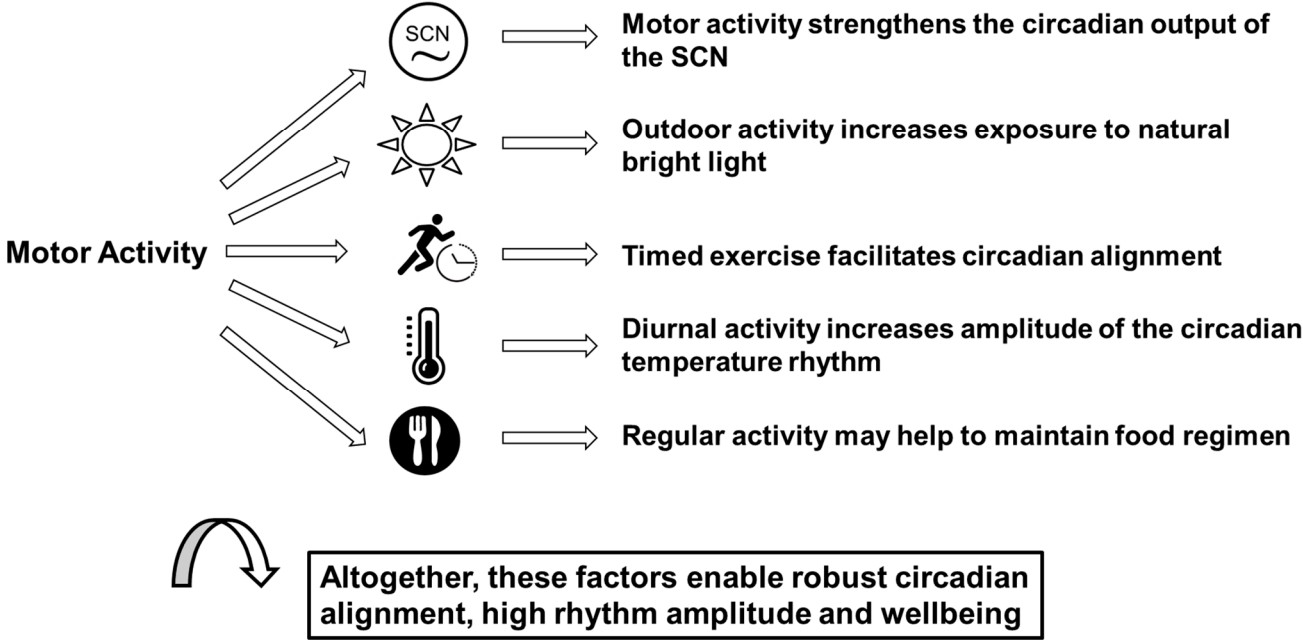

**Figure 1.** Daytime activity facilitates robust circadian rhythms and promotes health and well-being: schematic overview.

Voluntary exercise also strengthens the circadian system in elderly people. Thus, it can attenuate some age-related changes and improve their health and wellbeing. Higher levels of physical activity in older individuals correlates with a high circadian amplitude of clock gene Per3 expression [127]. Moreover, the rhythmic expression of PER3 is associated with physical fitness in older adults. Dupon Rocher et al. have shown that the amplitude of the circadian temperature rhythm is correlated with a higher level of physical activity in older adults [128].

The resynchronization time, e.g., following a time-zone transition, can be shortened, and the adaptation to shift work be improved by means of physical exercise. This has been shown particularly in experiments with induced jet lag [129,130]. Moreover, physical exercise is generally useful to adjust human circadian rhythms to external time cues [131,132] and to synchronize the whole circadian system [133].

As a consequence of stabilized circadian rhythms by means of enhanced physical activity, the cognitive performance can be improved. The SCN are involved in cognitive processes, and stable circadian rhythms are essential for cognitive abilities [134]. Disruption of the circadian sleep–wake cycle may have adverse consequences for neurobehavioral performance [135], because sleep is one of the most critical factors, which may affect cognitive abilities by improving the consolidation of information [136–138]. Moreover, processes such as learning and memory are modulated in a circadian manner.

The importance of the circadian system for cognitive processes has been shown particularly in animal experiments. First, the best cognitive performance was demonstrated in studies where the temporal sequence of tests was in harmony with the 24-h rhythm [139]. Second, cognitive performance differs over the 24-h scale, or in a circadian manner [140,141]. Third, the disruption of the circadian rhythms compromises cognitive abilities. In rats exposed to repeated phase shifts of the light–dark cycle, the learning in the Morris water-maze—particularly the long-term retention—was impaired [142]. Studies of Ruby et al. on arrhythmic Djungarian hamsters revealed substantial deficits in object recognition and spatial working memory [143–145]. These hamsters were rendered arrhythmic by light treatment [146]. In our colony of Djungarian hamsters, the arrhythmic phenotype emerges

spontaneously [98]. Djungarian hamsters with a well-functioning circadian system show a pronounced daily rhythm of cognitive abilities, with the best performance during the activity period. On the other hand, animals of the arrhythmic phenotype, in which the SCN do not generate a circadian signal, not only have no rhythm in their cognitive abilities, but they also fail to perform a novel object recognition task, and show an impaired individual recognition and social memory performance [7,141]. When the animals had access to a running wheel, part of them reestablished a circadian activity rhythm, namely those who used the wheel most intensively. In these animals, cognitive abilities were also restored [10].

Daytime activity associates with exposure to more bright ambient light, temperature circadian amplitude increase–factors that altogether foster circadian alignment and may help to maintain a healthy circadian diet.

### 3. Conclusions

In the modern world, lifestyle is responsible for almost two-thirds of all cases of disease globally. The World Health Organization has pinpointed the lack of exercise as one of the most important factors causing lifestyle diseases. Not only the amount of activity, but also the time of exercise is important, since time-scheduled activity can be a circadian zeitgeber.

**Author Contributions:** D.W.: original draft preparation, review and editing; D.G.: review and editing. All authors have read and agreed to the published version of the manuscript.

**Funding:** The study was supported by the West-Siberian Science and Education Center, Government of Tyumen District, Decree of 20 November 2020, No. 928-rp. (D.G.).

**Institutional Review Board Statement:** Ethical review and approval were waived for this study due to it being a comprehensive review of already published data.

**Informed Consent Statement:** Patients consent was waived as already published data were used.

**Data Availability Statement:** Not applicable.

**Conflicts of Interest:** The authors declare no conflict of interest.

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
