# Peer review of "The Impact of Physical Activity on the Circadian System: Benefits for Health, Performance and Wellbeing"

_applsci, doi:10.3390/app12189220_

Round 1

Reviewer 1 Report

This study conducted a review to investigate the impact of physical activity on the circadian system,

benefits for health, performance and wellbeing. The quality of the manuscript is unsatisfactory, reflected by the following issues.

1.       The abstract is too long to read. Also, the aims and objectives of the study were missing.

2.       The introduction failed to state the novelty and significance of the study.

3.       There should have been a method part to show how to conduct the review.

4.       After the review, the authors should have provided the research limitations of the reviewed studies and future research agenda.

5.       What are the practical implications of the study?

Author Response

Response: We thank the reviewer for the comments.

  1. The abstract is too long to read. Also, the aims and objectives of the study were missing.

Response: We read carefully this journal’s recommendations for Review papers. The aim of the reviews is to offer a comprehensive analysis of the extant literature within a field of study as it mentioned in the abstract (P.2; L.37). The Abstract was shortened.

  1. The introduction failed to state the novelty and significance of the study.

Response: Since we prepared a review and not an original paper, these points seem to be not relevant.

3.       There should have been a method part to show how to conduct the review.

Response: As we prepared a review and not an original paper, these points seem to be not relevant.

  1. After the review, the authors should have provided the research limitations of the reviewed studies and future research agenda.

Response: The following text was added, P.6, L.223:

This review is based mainly on animal studies, limiting immediate translation and application of the findings to humans. Therefore, future research aimed to clarify benefits of timed physical activity in men and women depending on age, chronotype and relevant genetic polymorphisms are prompted.

Benefits have been shown also for humans. However, the underlying mechanisms can be investigated only in animals.

5.       What are the practical implications of the study?

Response: Featured application at the beginning (P1., L.13) was modified to stress practical implications: “Review summarizes current knowledge regarding the effects of physical activity on the robustness of the circadian system and the benefits for health, performance, and wellbeing. We propagate practical implications of circadian activity rhythms for diagnosis and the advantages of scheduled motor activity.

Reviewer 2 Report

The review, “The Impact of Physical Activity on the Circadian System. Benefits for Health, Performance, and Wellbeing,” by Weinert and Gubin is well-written. The authors have cited many recently published articles relevant to the subject. However, there are a few issues related to the language. I have pointed out below some issues that I noticed in the abstract of the review. Similar problems are there in the entire text of the review.

Line #18: Delete ‘s’ from ‘guarantees’ and replace ‘an’ with ‘the.’

Line #22: Delete ‘an’ placed before ‘adequate.’

Lines #23-25: The phrase, “It has not only … the main circadian pacemaker,” could be rephrased like this: “It has not only generally favorable effects on the cardiovascular system, energy metabolism, and mental health, for example, it may also stabilize the circadian system via feedback effects on the SCN, the main circadian pacemaker.”

Line #28: The phrase, ‘It promotes the entrainment to external periodicities’ needs to be rephrased. Entrainment of what? ‘Entrainment to’ or ‘Entrainment of?’

Line #42: ‘aim or, should be ‘aim of.’

Figure 1: I found the following issues in the figure.

Outdoor activity increases exposure no natural bright light. In the preceding ‘exposure no’ should be ‘exposure to.’

“Timed activity facilitates circadian alignment.” Is it a timed exercise?

Author Response

The review, “The Impact of Physical Activity on the Circadian System. Benefits for Health, Performance, and Wellbeing,” by Weinert and Gubin is well-written. The authors have cited many recently published articles relevant to the subject. However, there are a few issues related to the language. I have pointed out below some issues that I noticed in the abstract of the review. Similar problems are there in the entire text of the review.

Response: We are thankful to the reviewer for this important comment. The text was thoroughly proofread.

Line #18: Delete ‘s’ from ‘guarantees’ and replace ‘an’ with ‘the.’

Response: done

Line #22: Delete ‘an’ placed before ‘adequate.’

Response: done

Lines #23-25: The phrase, “It has not only … the main circadian pacemaker,” could be rephrased like this: “It has not only generally favorable effects on the cardiovascular system, energy metabolism, and mental health, for example, it may also stabilize the circadian system via feedback effects on the SCN, the main circadian pacemaker.”

Response: We can´t see the difference. The referee did delete “the” before “energy” and combined two phrases into one (replaced a full stop by a comma). Also, according to suggestions from reviewer 1 this part of abstract was rephrased to be shortened.

Line #28: The phrase, ‘It promotes the entrainment to external periodicities’ needs to be rephrased. Entrainment of what? ‘Entrainment to’ or ‘Entrainment of?’

Response: Text was accordingly changed to "promotes their entrainment"

Line #42: ‘aim or, should be ‘aim of.’

Response: Is now corrected.

Figure 1: I found the following issues in the figure.

Outdoor activity increases exposure no natural bright light. In the preceding ‘exposure no’ should be ‘exposure to.’

Timed activity facilitates circadian alignment.” Is it a timed exercise?

Response: We have changed it accordingly (exposure to, exercise instead of activity).

Reviewer 3 Report

As described by the authors, “The aim of the present paper is to summarize the current knowledge concerning the effects of physical activity on the stability of the circadian system and the corresponding consequences for physical and mental performance.”

In accordance with this, the following items are discussed. A sufficient number of papers are cited to provide a meaningful discussion.

#How does exercise affect circadian rhythms and through what pathways does it control them?

#What effects do the stabilization of circadian rhythms have on the body?

What problems arise when circadian rhythms are disrupted?

There are several concerns.

#Social jet lag is one of the negative effects on the body. Please discuss social jet lag, what it is and what the problems are.

#The effects of exercise on the body are out of scope here and should not be stated.

#There are many grammatical errors; English proofreading is needed.

Author Response

As described by the authors, “The aim of the present paper is to summarize the current knowledge concerning the effects of physical activity on the stability of the circadian system and the corresponding consequences for physical and mental performance.”

In accordance with this, the following items are discussed. A sufficient number of papers are cited to provide a meaningful discussion.

#How does exercise affect circadian rhythms and through what pathways does it control them?

#What effects do the stabilization of circadian rhythms have on the body?

#What problems arise when circadian rhythms are disrupted?

Response: We are thankful to the reviewer for this important comment. As we understand, these are questions we have sufficiently discussed in the paper, i.e., there is no need to respond.

There are several concerns.

#Social jet lag is one of the negative effects on the body. Please discuss social jet lag, what it is and what the problems are.

Response: "Social jetlag" has of course a negative impact, though this is behind the scope of the paper.

#The effects of exercise on the body are out of scope here and should not be stated. 

Response: As far as we understand, the referee means the chapter 2.2.. We believe, this important as many of these effects are mediated via the circadian system.

#There are many grammatical errors; English proofreading is needed.

Response: The ms was checked thoroughly. Pro-Writing Aid software was used.

Round 2

Reviewer 1 Report

The authors failed to address my previous comments. For example, they failed to provide the details of how to conduct the review. Also, the practical implications are not specific. The novelty and significance of this work are low.